# Thermoplastic Polymeric Materials for Spacecraft Applications: Flame Retardant Properties and UV/AtOx Aging Analysis

**Roberto Pastore [1,\*], Marta Albano [2], Andrea Delfini [1], Fabio Santoni [1] and Mario Marchetti [1]**

[1] Astronautics, Electric and Energy Engineering Department, Sapienza University of Rome, via Eudossiana 18, 00184 Rome, Italy; andrea.delfini@uniroma1.it (A.D.); fabio.santoni@uniroma1.it (F.S.); mario.marchetti@uniroma1.it (M.M.)

[2] Italian Space Agency (ASI), Via del Politecnico snc, 00133 Rome, Italy; marta.albano@asi.it

[\*] Correspondence: roberto.pastore@uniroma1.it

**Abstract:** Space-mission development introduced the problem of human isolation in extreme environments. The integration of architectural concepts such as windows, with their technical implications into human space missions, is necessary especially for long-duration space flights. Such solutions must be subjected to close certification testing in order to establish their compliance with severe space environment conditions. Moreover, projects of long-term missions involve a renewed concern about fire safety in manned space vehicles. The supposed occupancy time of the order of decades, in fact, makes unplanned fire ignition events virtually unavoidable. An experimental test-plan performed to qualify a commercial thermoplastic material for applications as transparent element in spacecraft is reported in the present work. A flame exposure test, as well as ultraviolet radiation and atomic oxygen erosion rate measurements, was carried out on a commercial poly-sulfone material, following ASTM standard procedures. The aim of the research was to develop and put forward flame retardant structures able to withstand the harsh space environment, preventing or mitigating the degradation of their physical and chemical integrity, with particular regard to the visible-light transparency. The results obtained show that the tested poly-sulfone may be considered as a promising material for the claimed application, even if further surface optimization treatments should be conceived in order to gain a full adaptability to the operative constraints.

**Keywords:** space environment; flame retardants; ultraviolet aging; atomic oxygen erosion; thermoplastic poly-sulfone

## 1. Introduction

The operational spacecraft environment of low earth orbit (LEO) is characterized by the damaging synergic effects of solar radiations, such as ultraviolet and X-rays, atomic oxygen impact and ionizing radiations; thus, protecting from these environmental conditions represents a fundamental task for every space mission [1–6]. Furthermore, the flammability screening test is one element of a rigorously applied set of measures designed to minimize the hazard of fire in manned spacecraft, thus becoming crucial as it determines, in large measure, the typology of controls imposed on a material as potential fire threat [7–9].

Considering a normal habitat on the Earth, a very common element of the living built environment, with basic importance from a psychological point of view, is the window. The ESA "FlexWin" (Flexible Window for inflatable habitats) funded project was devoted to investigate innovative materials and solutions to be used as optical parts, with particular reference to the inflatable modules [10]. The need to achieve a lightweight concept, able to couple light transmission, mechanical resistance and integration with the interfaces of inflatable module's functional layers, led to the selection of the family of engineering polymers—in particular, poly-sulfones (PSUs)—as material candidates [11]. PSUs are light transparent thermoplastic polymers usually employed for their toughness, impact strength

and flame resistance [12]. Such characteristics could make them competitive with respect to the glass-based materials so far employed for spacecraft optical components.

In this work, the experimental testing activity carried out in order to qualify a commercial thermoplastic sulfone polymeric material is reported, including flammability test, as well as ultraviolet (UV) and atomic oxygen (AtOx) erosion analysis. The former evaluates the response to a well-defined laminar flame at the bottom of a test sample and is currently used to screen for flammability all materials intended for using inside manned spacecraft [8]. The latter evaluates the materials behavior in LEO working conditions, focusing in particular on the surfaces aging due to the combined effects induced by high vacuum and UV/AtOx irradiation [2,13–20]. The results reported in the paper show that applying the proposed solution in spacecraft design could provide overall architectural and functional advantages, as far as the LCC reduction is concerned. The materials used, in fact, are generally low-price and available, which does not translate into sharp increases in the price of the potential product. Such a feature is particularly suitable in the view of the currently adopted concepts driving aerospace industry development; nowadays space economy and commercialization demand for multi-functional structures, as well as time/cost-saving procedures, with particular emphasis on low-waste technological processing and components reusability and biocompatibility.

## 2. Materials and Methods

### 2.1. Thermoplastic Material

The analyzed samples are made of the product Radel R, provided from Solvay [21]. Radel belongs to sulfone polymers family (chemical compounds containing sulfonyl functional groups attached to two carbon atoms): It can be supplied in two chemical forms, as polyphenylsulfone (PPSU—R type) or polyethersulfone (PES—A type). Such material can withstand continuous exposure to moisture and high temperatures, and may absorb tremendous impacts without cracking or breaking; other important Radel properties are transparency and flame-retardant capability. Some of the Radel samples were coated by using a hybrid sinterable polyphenylene-based sulfone (PPSO2), purchased at Ceramer [22]; such treatment is intended to achieve a surface with improved tribological properties, like wear and abrasion resistance.

### 2.2. Flammability Experimental Setup

The fire testing was carried out by means of a flammability test facility (FTF) in compliance with the ASTM D6413 international standard. The FTF, fabricated by Govmark Testing Services, was realized in a ventilate stainless steel chamber (see Figure 1). Inside the chamber, a stationary burner with a gas pilot tube is allocated. The flame is adjustable by two gas inlet controlling valves (one on the methane gas cylinder and the other outside the chamber). A flame controller was placed on the left side of the cabinet, with a timer for the measurement of the flame time, the after-flame and after-glow time (see definitions below). The evacuation flow rate was 2 $m^3$/h, and the chamber was placed in a hood in order to evacuate eventual burning smokes after the test execution. The samples subjected to the flammability test were 6 mm–thick rectangular slabs (dimensions: $300 \times 64$ $mm^2$); one test series was carried out on two-layered samples, to assess the influence of thickness. The test method approached the procedure of ECSS-Q-70-21A standard [8]. The samples were conditioned in a clean room, at 23 °C, for at least 24 h; then the test was performed within 4 min after the removal of the sample from the clean room. Flame height and position (centered at the bottom edge of the tester) were adjusted before the test (i.e., without the sample), and the pilot flame was left lighted. The sample was then positioned vertically to the flame. The purpose of the test is to determine the material's flammability characteristics by the exposure to a flame applied at the bottom edge of the material samples [9,23–28]. The flame-time was currently set to 12 s for different materials comparison; the reference thermoplastic was also tested at several flaming time intervals, in order to gain further knowledge about the fire-retardant effective dynamics. The testing recorded temperature

(measured by K-thermocouples) was around 450 °C. The main measured parameters were as follows:

- Char length—distance from the sample edge directly exposed to the flame up to the furthest point of visible damaging;
- After-flame time—time of sample flaming after the ignition source has been removed;
- After-glow time—time of sample glowing after ignition source removal and flame extinction.

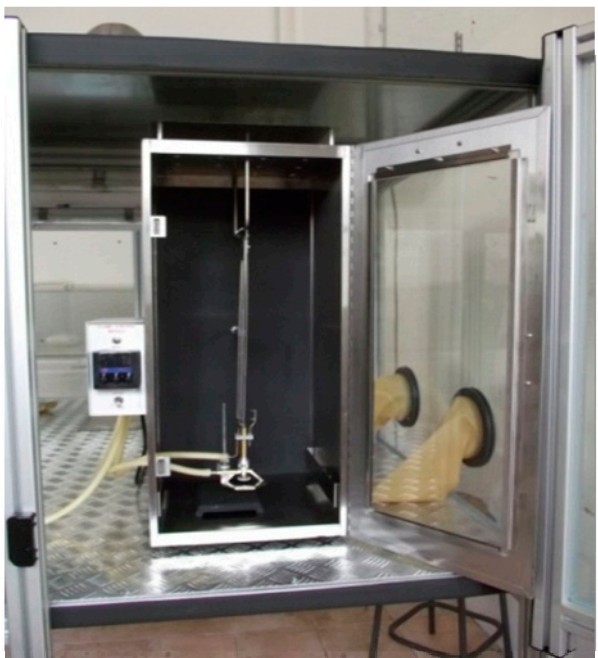

**Figure 1.** Picture of the flammability test facility.

The standard typically adopted for acceptance or rejection is that the material has to be non-combustible or self-extinguishing. That means that the combustion zone has to propagate less than one half of the height into the sample (150 mm in this case) and that the burning time has to not exceed 10 min. Moreover, there should be no sparking, sputtering or dripping of flaming particles from the test sample. Finally, all the samples (three is the minimum number required) must show no failure, in order to fulfill the standard requirements.

### 2.3. UV/AtOx Exposure Experimental Setup

The adopted UV/AtOx test facility simulates the ultraviolet irradiation and the atomic oxygen impact effect in low orbit space environment [16,19,20]. It is constituted by a small chamber (see Figure 2) equipped with a rotary vane pump and a Varian V-550 turbo-molecular pump in order to set ultra-high vacuum conditions (minimum pressure achievable: $10^{-5}$ Pa). The aim of the UV/AtOx test is the evaluation of the material degradation in terms of mass loss due to erosion: The samples are weighted before and after the test, using a Mettler-Toledo XP26DR microbalance (precision: 1 μg). The tests are performed in a clean room conditioned at room temperature for 48 h, at a mean pressure of $2.6 \times 10^{-1}$ mbar, in order to avoid weight errors due to moisture losses.

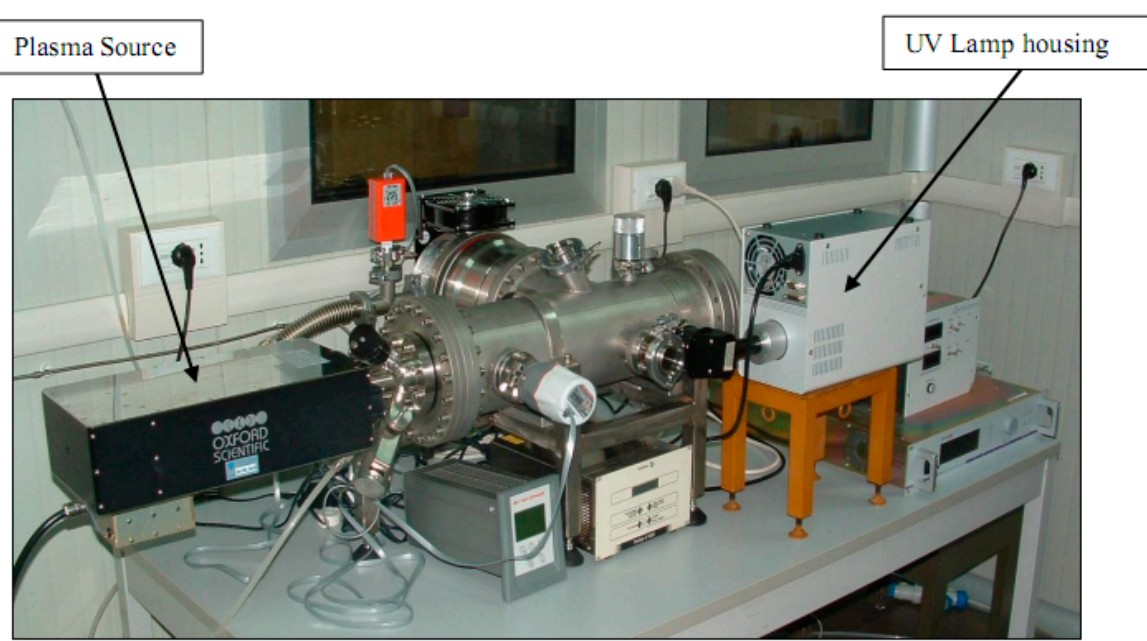

**Figure 2.** Picture of the UV/AtOx simulator test facility.

The UV generator is a highly stable mercury–xenon lamp produced by Hamamatsu. The system has high intensity UV line spectra with an elliptical reflector (UV cold mirror) having reflectivity higher than 90% in the UV range, and a quartz light guide with UV transmittance. The lamp works in horizontal position in order to have an optical system with low light loss. The spectral emittance field range is 200–600 nm, with maximum emission at 365 nm. The radiation intensity of the lamp system is 410 mW/cm$^2$ (around 10 Suns) at a distance of 60 mm, with an aperture size of 20 mm.

The AtOx flux is simulated by generating an oxygen plasma, using an OS-PREY Plasma Source, which is a radio-frequency inductively coupled plasma source produced by Oxford Scientific (SPECS) Instruments. The operating principle is based on the ionization of the oxygen molecules by coupling the energy from a radio-frequency power source (13.56 MHz) to an ionized gas. The source generates a beam constituted by 99% of neutral elements (around 60% of O atoms and 40% of O$_2$ molecules) and 1% of O$^+$ ions with energy $5 \div 25$ eV. The sample distance from the quartz emission aperture at a right angle orientation respect to the beam is of 50 mm, as it is the worst colliding condition. The plasma flux is calculated by using a reference sample of Kapton HN, a material of known in-space erosion yield, as suggested by ASTM E2089. The samples are exposed to AtOx at an effective fluence (flux over time) of $1.25 \times 10^{20}$ atom/cm$^2$; based on the measured mass loss, the erosion rate is calculated from the sample density, the sample exposed area and the produced fluence.

## 3. Results and Discussion

### 3.1. Flammability Test

Three typologies of Radel R specimen series were tested: as-received (R), double layered (R-2L) and Ceramer-coated (R-HY). The as-received material was also tested for several flame exposition times; in Table 1, the characteristics of the different sets of measurement are specified, including the double duration test (R-2T). The time/length measurement results are also reported: Such values represent averaged results by testing at least three samples for each class. The char length is measured after cleaning operations needed to remove the smoke deposition onto the samples surface. Besides the aforementioned parameters (char length and after-flame/glow times), further information is provided, such as the smoke length (measured before the sample cleaning), the damaged length (referred to the part of material which is visibly damaged but not charred) and the char width (the

length of the base line of the charred area, at the bottom edge of the sample). Moreover, the test chamber is thoroughly analyzed after each test, in order to check if the specimen sputtered some particles due to the flame exposure (material spread).

**Table 1.** Flammability test: main characteristics of samples classes and experimental setup (upper box), and experimental results (lower box).

| Sample Series | R | R-2L | R-HY | R-2T |
|---|---|---|---|---|
| Material | Radel R | Radel R double layer | Ceramer-coated Radel R | Radel R |
| # of samples | 4 | 3 | 3 | 4 |
| dimensions (mm) | 300 × 64 × 6 | 300 × 64 × 12 | 300 × 64 × 6 * | 300 × 64 × 6 |
| flame time (s) | 12 | 12 | 12 | 24 |
| smokes length (mm) | 163 | 70 | 208 | 225 |
| damage length (mm) | 11 | 17 | 42 | 26 |
| char length (mm) | 4.5 | 8.4 | 4.6 | 11.2 |
| char width (mm) | 24.4 | 30.0 | 21.5 | 15.1 |
| after flame time (s) | 1.1 | 1.4 | 1.1 | 0.9 |
| after glow time (s) | no glow | no glow | no glow | no glow |
| material spread | N | N | Y | N |

* The thin thickness of the coating is not taken into account. R, as-received Radel R; R-2L, double-layered Radel R; R-HY, Ceramer-coated Radel R; R-2T, Radel R double duration test.

Some pictures of the tested samples are reported in Figure 3. All the uncoated Radel samples showed no glow and did not spread visible amount of material during the test. The exposure to the flame gradually damaged the materials: A thin line distinguishing the contour of the charred area is clearly visible. The char length is generally not much extended, but there is an area where matter melting occurred. The obtained findings are in good agreement with results reported in the literature for this kind of thermoplastic material (see, in particular, Table 3 in Reference [12]. The damaged length describes such phenomena: In the melted area, the materials still show a visible transparency; this area is smooth and it is not detectable by handle touch. The charred area is more evident at the external sides of the slabs: It is irregular, and little bulges are detected, likely due to gas releasing during combustion (Figure 3a).

The behavior of the double-layered samples (R-2L) is more effective than what observed for the reference R class: The part of the samples between the two layers seems to be not affected by the flame, and the bottom edge of the samples shows non-uniform burns, while only the external sides of the samples are melted and charred (Figure 3b). The behavior of the coated Radel samples (R-HY) revealed some differences in respect to the other tests. Smokes are not uniform, and the damaged area is not marked by a clear boundary between melted area and undamaged part (Figure 3c): It is composed of a first charred area with some bulges, a second charred area without bulges, and a white-yellow area just around, related to the coating degradation. The charred area without bulges can be removed by means of abrasive cloth, since the friction of simple paper tissues (used to remove smokes) is useless. Moreover, it was observed that the R-HY specimens spread visible particles of material around the final seconds of the flame test, while, at the end of each test, a little quantity of ash was discovered; the spread material could be likely identified with the hybrid coating. In the longer flame exposure time tests (R-2T), some traces of increasing damaging were founded. In particular, the higher char length and the presence of darker zones of shape similar to that of the smoke areas (that is, the inner trace of the smokes) indicates the worsening of the Radel R when exposed to flame for longer times (Figure 3d). Anyway, it has to be stressed that all the R-2T specimens have shown to be self-extinguishing and that no melted material dripped off the samples.

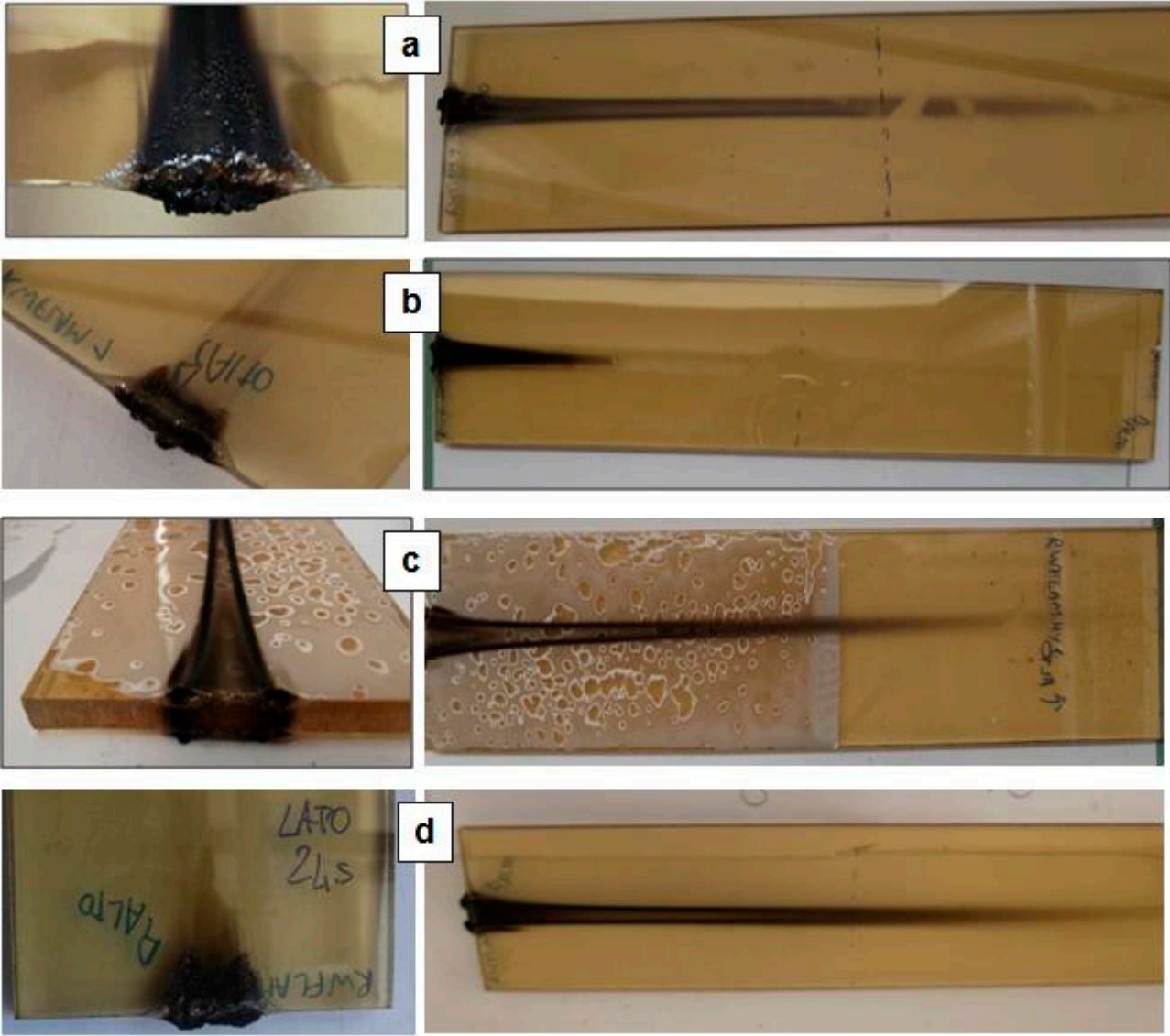

**Figure 3.** Flammability test: pictures of specimen charred area (left) and damaged surface (right) from (**a**) R, (**b**) R-2L, (**c**) R-HY and (**d**) R-2T testing series.

The measured smoke lengths indicate that only R-2L test series satisfied the abovementioned condition of damaging propagation limited to one half of sample height (150 mm) for the scheduled time of flame exposure (12 s). The correspondent results obtained by analyzing the base PSU in shorter and longer tests allow us to gain insights about the damage evolution, in order to estimate an optimal thickness and envisage the material performance for prolonged flame exposures. Experimental data of smoke length measured at four/two flame-times for Radel R/R-2L specimens are reported in Figure 4, where the curves estimating the effective behavior of single and double-layered material are obtained by simple physical modeling. A generic power law regression based on a functional behavior as

$$l(t) \propto t^\alpha \tag{1}$$

leads to an optimal value of the exponent $\alpha \sim 0.52$, which meaningfully suggests an effective quadratic dependence between the variables. Such a finding can be easily retrieved by assuming a schematic framework where the damaging progression $\Delta l(t)$ may be expressed as follows:

$$\Delta l = \frac{k}{l \cdot S^2} \Delta t \tag{2}$$

where the parameter $k$ is characteristic of the specific material, having considered that the damage forward is doubly retarded by the surface ($S$) heat dissipation and due to the distance from the burning point. By imposing trivial initial conditions, the differential relationship (2) leads to the functional expression:

$$l(t) = \frac{1}{S}\sqrt{2k{\cdot}t} \tag{3}$$

which is in agreement with the above considerations and is used to fit the experimental data, giving an optimal value for the coefficient $k \sim 1.54 \times 10^{-10}\ \mathrm{m^6/s}$. In such a framework, it can be thus established that a slab specimen of the analyzed PSU material should fulfill the damaging propagation constraint with a thickness around 6.5 mm, while the double-layered component is believed to provide flame retardant effective behavior up and beyond 40 s of fire exposition, as far as the adopted setup is concerned.

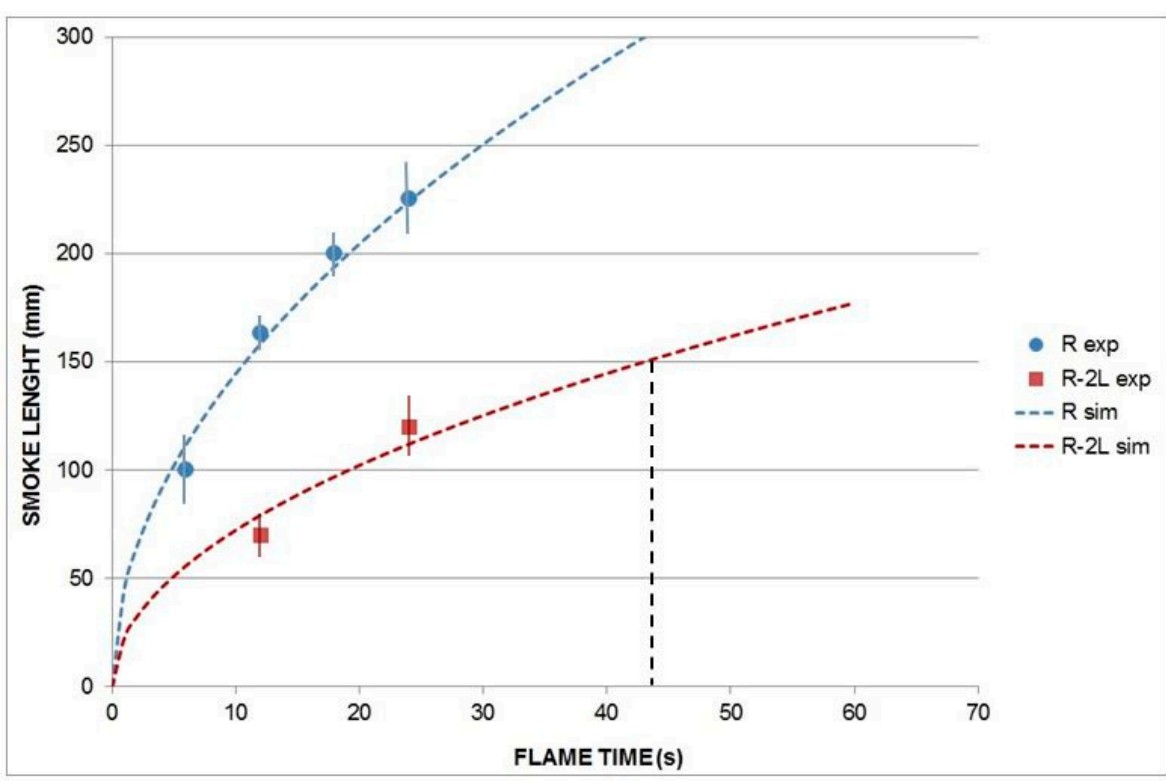

**Figure 4.** Flammability test: smoke length versus flame-time for Radel R material, experimental data and physical optimized modeling results.

To summarize, Radel R material showed to be non-combustible, well resistant to flame exposure and to keep visible transparency. As expected, the thicker the material slab, the lower the flame damaging, whereas a sharper degradation occurs for prolonged exposures. Thus, this kind of thermoplastic material provides an acceptable performance within the scenario of polymeric flame-retardant solutions. On the other hand, the hybrid coating seems to worsen the Radel properties, with particles emission during the melting, thus resulting in a non-admissible material for anti-flame applications.

### 3.2. UV and Atomic Oxygen Test

UV and AtOx aging characterization was carried out on the pure Radel R and on the hybrid coated R-HY. The effect of AtOx alone was first analyzed, and then the specimens were subjected to the blending UV+AtOx flux, as an attempt to discriminate different degradation phenomena and to appreciate the synergic erosion due to the combined action of two aging factors. Six samples (dimensions: $20 \times 20 \times 6\ \mathrm{cm^3}$) for each kind of material

were analyzed in all the tests; the corresponding results about the erosion rate $E_Y$ are given, by average, in Figure 5, recalling that this parameters is calculated from the measured mass loss, $\Delta M$, by using the following relationship:

$$E_Y = \frac{\Delta M}{\rho \cdot A \cdot F} \tag{4}$$

where $\rho$ is the material density (~1.33 g/cm$^3$), $A$ is the sample exposed area (~4.20 cm$^2$) and $F$ is the effective fluence obtained within the chamber (~$1.25 \times 10^{20}$ atom/cm$^2$). The results obtained clearly show the ineffectiveness of the ceramic coating treatment, which gives rise to a sharp increase of the erosion rate in both AtOx and UV+AtOx exposure testing with respect to the naked material. Rather than to low anti-oxidation properties of the ceramic itself, such behavior has to be related to a weak adhesion to the base material, as well as to the presence of defects, pores and scratches on the coated surface. On the contrary, the Radel R itself demonstrates a good resistance to both AtOx and UV+AtOx irradiation. In particular, only a little increase of about $6 \div 8\%$ of erosion is detected in the synergic conditions.

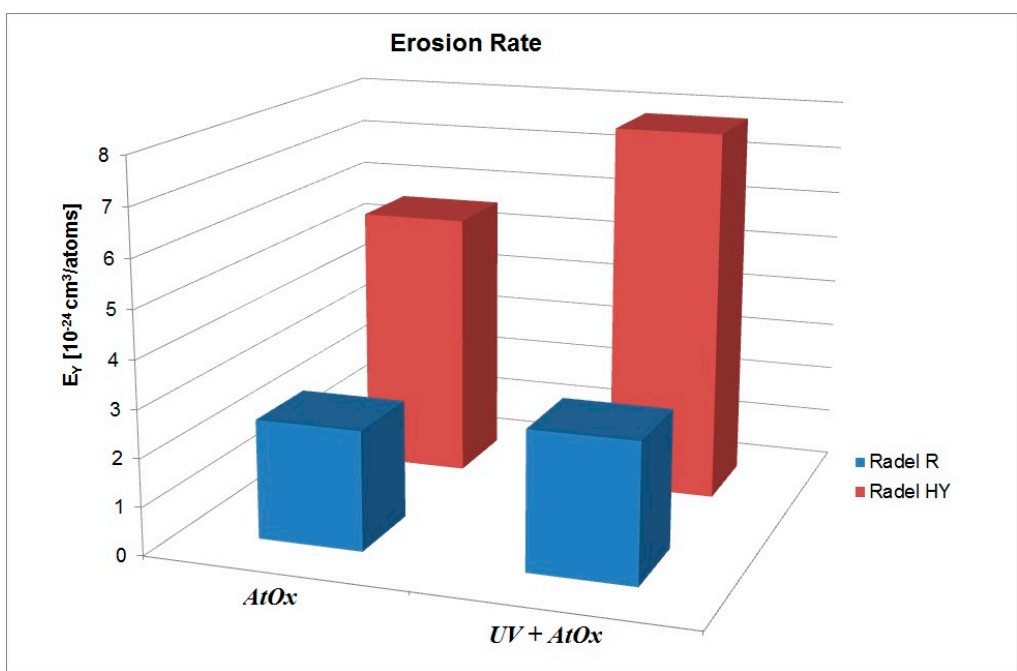

**Figure 5.** UV/AtOx aging characterization: erosion rate estimated from the measured mass loss.

The effect of UV exposure on the Radel R material is evident by the comparison between the appearance of the surface before and after the test (Figure 6). It is noticeable that the sample surface changed from light to dark yellow; thus, the transparency is, even though not dramatically, quite affected by UV radiation. That is the so-called "darkening effect", due to UV photon energy high enough to cause surface molecule activation and organic bond breakage, resulting in the degradation of the polymer optical properties [14,29]. After AtOx alone exposure, on the contrary, the samples' transparency appeared almost unchanged, with no evident changes of the surface morphology. The erosion yield obtained in the adopted experimental setup is in good agreement to the results obtained for poly-sulfone materials in the most of other ground-based facilities, as well as in space-flight experiments [2,13,15]. The low surface recession shown by Radel R testified the good anti-oxidation properties of poly-sulfone materials, which are supposed to be due to residue formation after the exposure to LEO environment, acting as a barrier layer that slows down the material erosion. Such capability, on the other hand, is strictly

related to the worsening of the material optical properties, as, for example, the degradation of the polymer original transparency.

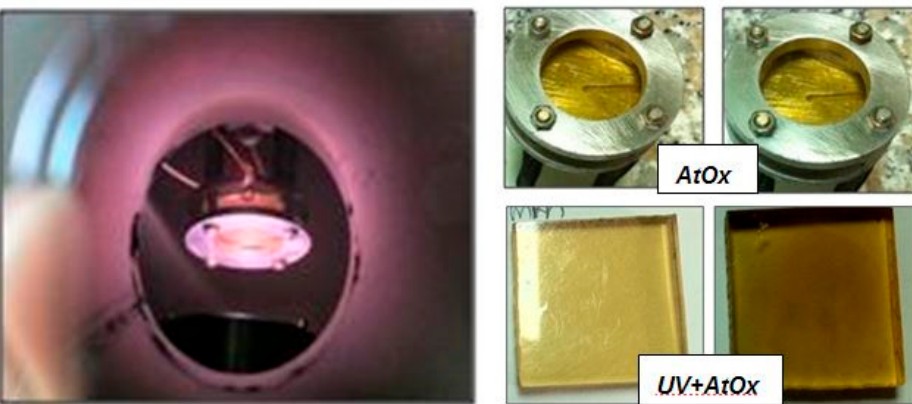

**Figure 6.** UV/AtOx aging characterization: chamber inside during the test, and Radel R samples before and after exposure, for both AtOx alone and UV+AtOx irradiation.

## 4. Conclusions

Long-duration space flights imply the integration of architectural concepts into human space missions, with the introduction of common elements of the living environment, such as windows, within the spacecraft design. The proposed solutions, of course, must be subjected to close certification, in order to establish their compliance with the severe space environment conditions and requirements. Light, transparent thermoplastic polymers such as poly-sulfones are believed to be ideal candidates fur such applications, thanks to their significant properties of toughness, impact strength and flame-retardant capability. The commercial poly-sulfone Radel R was characterized by flame exposure test, as well as by ultraviolet and atomic oxygen erosion rate measurements, in order to validate such material as thermoplastic polymer for space employment. Flammability testing showed that Radel R is self-extinguishing, with no afterglow phenomena, thus establishing the optimal behavior of poly-sulfones for anti-flame applications. Mainly, except for small charred areas, the melted zones appeared to keep visible transparency after the flame exposure, thus allowing us to assess the material suitability for the required applications. The results of UV and AtOx erosion test demonstrated a suitable chemical stability of Radel R, even if further optimization treatments on the base poly-sulfone surface should be performed to achieve optimal results. In fact, the hitting mechanism by UV radiations gives rise to a significant interaction with the impinged polymer surface, resulting in a lowering of the polymer transparency. Even if not relevant when considered alone, the AtOx effects result in being more damaging due to the synergistic effect with UV radiation, which may cause more severe erosion of spacecraft materials, thus compromising their optical properties. Moreover, AtOx corrosion effects should be closely considered in terms of flammability index lowering, which would make simultaneous UV exposure able to light the spark of fire ignition events. The proposed coating treatment performed by using the commercial hybrid ceramic polymer Ceramer appeared to worsen the surface properties of the base poly-sulfone Radel R. In particular, a significant amount of material spreads during the flammability test and there is a sharp increase of UV/AtOx aging, thus establishing the unsuitability of such kind of poly-sulfone coating for space applications. Further surface optimization treatment should be developed in order to gain a fully compliance with the operative requirements. By this way, the next activities of the research project will be focused on the development of a coating process able to smooth the optical properties degradation due to LEO environment effects, while keeping unchanged (or eventually improving) the anti-flame effectiveness of the base poly-sulfone material. Future research will be addressed also to take into account thermal properties over as wide a temperature range as possible, as well as the evaluation of the depth of transparency

degradation as function of UV exposure time. Finally, it is worth noting that, since this is a ground-based study, the comparisons are inevitably valid only for normal gravity: A definitive assessment of the applicability of any of these test methods to a micro-gravity environment awaits a quantitative interpretation of the role of gravity, especially as far as the flammability is concerned.

**Author Contributions:** Conceptualization, M.M.; Methodology, M.M. and F.S.; Software, R.P.; Validation, M.A. and A.D.; Formal Analysis, R.P., M.A. and A.D.; Investigation, M.A. and A.D.; Data Curation, R.P.; Writing—Original Draft Preparation, R.P., M.A. and A.D.; Writing—Review & Editing, R.P.; Supervision, M.M. and F.S. All authors have read and agreed to the published version of the manuscript.

**Funding:** This research received no external funding.

**Institutional Review Board Statement:** Not applicable.

**Informed Consent Statement:** Not applicable.

**Data Availability Statement:** Not applicable.

**Conflicts of Interest:** The authors declare no conflict of interest.

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
