# Peer review of "Thermoplastic Polymeric Materials for Spacecraft Applications: Flame Retardant Properties and UV/AtOx Aging Analysis"

_applsci, doi:10.3390/app11030949_

Round 1

Reviewer 1 Report

The material physical/chemical and mechanical properties characterization when submitted to extreme conditions is currently a hot topic in polymeric material industry agenda, in this particular case applied to spacecraft design (one of the strengths of the manuscript). Nonetheless, the subject addressed is also applied to other kind of applications. The work goal is well addressed and adequately framed, and the research is presented in a transparent and comprehensive way. The manuscript is well structured and well written. The major shortcomings that I identify in the paper is the lack of:

  • Appropriate linkage to other works in the literature and, consequently, a proper results comparison;
  • Reference to a more recent works in the literature | Globally bibliography references are no current.

The paper will much benefit if a comparison between main results and results achieved by other research works is conducted. In addition, more recent works should be used to improve globally the manuscript.

Check:

  • Figure 5 formatting;
  • References formatting.

Author Response

Thanks a lot for Your valuable comments.

Appropriate link to inherent literature results was stressed at lines 165-166; a number of more recent references were also added to the citations list.

All the figures formatting was checked, and a deal of effort was done to put the references in the style required by App. Sci.

Reviewer 2 Report

The manuscript under review deals with an investigation on the usage of polisulfon based thermoplastic materials for Spacecraft applications. The studied materials were analyzed in: flammability tests and UV/AtOx aging tests. The research material and the obtained results were clearly described.

The manuscript is written in an interesting way and presents new data that may be useful not only in spacecraft research but also in other fields of science and applications, because polysulfones can be additives, for example, to optical fibers.

I believe it should be accepted for publication in Applied Sciences after some minor revision.

Line 130 witness sample – I suggest change e.g. for reference sample

Page 9 please check the position of the figures and their numbering and labels

It also would be good to include information whether the changes in transparency concerned the entire volume of the sample or only the surface exposed to radiation, and if so, to what depth the degradation took place.

Some spelling and editing errors need to be corrected.

Final remark - future research should take into account thermal properties over as wide a temperature range as possible.

Author Response

Thanks a lot for Your valuable comments.

- at line 130 the word ‘witness’ was replace by ‘reference’;

- position, numbering and label were checked at pag.9;

- no investigations were carried out on the depth of degradation due to UV exposure: it will be focus of the next experimental plan, as addedin the conclusions – thanks a lot for the hint;

- a deal of effort was done to correct spelling and editing errors;

- as added in the conclusions, future research will be addressed to take into account thermal properties over as wide a temperature range as possible – thanks again for the right suggestion.